# Advances of the Holographic Technique to Test the Basic Properties of the Thin-Film Organics: Refractivity Change and Novel Mechanism of the Nonlinear Attenuation Prediction

**DOI:** 10.3390/polym16182645

**Published:** 2024-09-19

**Authors:** Natalia Kamanina

**Affiliations:** 1Joint Stock Company Scientific and Production Corporation S.I. Vavilov State Optical Institute, Babushkina Str. dom 36, Korpus 1, St. Petersburg 192171, Russia; nvkamanina@mail.ru; Tel.: +7-812-327-0095; 2Vavilov State Optical Institute, Kadetskaya Liniya V.O., dom 5, Korpus 2, St. Petersburg 199053, Russia; 3Photonics Department, Electronics Faculty, St. Petersburg Electrotechnical University (“LETI”), Ul. Prof. Popova 5, St. Petersburg 197376, Russia; 4Department of Advanced Development, Petersburg Nuclear Physics Institute, National Research Center “Kurchatov Institute”, 1 md. Orlova Roshcha, Gatchina 188300, Russia

**Keywords:** organic structures, polyimides, COANP, NPP, PNP, carbon nanoparticles and tubes, laser-matter interaction, refractivity change, intermolecular charge transfer complex formation

## Abstract

A large number of the thin-film organic structures (polyimides, 2-cyclooctylarnino-5-nitropyridine, N-(4-nitrophenyl)-(L)-prolinol, 2-(n-Prolinol)-5-nitropyridine) sensitized with the different types of the nano-objects (fullerenes, carbon nanotubes, quantum dots, shungites, reduced graphene oxides) are presented, which are studied using the holographic technique under the Raman–Nath diffraction conditions. Pulsed laser irradiation testing of these materials predicts a dramatic increase of the laser-induced refractive index, which is in several orders of the magnitude greater compared to pure materials. The estimated nonlinear refraction coefficients and the cubic nonlinearities for the materials studied are close to or larger than those known for volumetric inorganic crystals. The role of the intermolecular charge transfer complex formation is considered as the essential in the refractivity increase in nano-objects-doped organics. As a new idea, the shift of charge from the intramolecular donor fragment to the intermolecular acceptors can be proposed as the development of Janus particles. The energy losses via diffraction are considered as an additional mechanism to explain the nonlinear attenuation of the laser beam.

## 1. Introduction

It is well known that polymers are widely used as pure matrices and composites, whose properties can be dramatically changed due to the introduction of various kinds of sensitizers, both inorganic and organic. This modification of doped polymer composites permits an increase in the charge carrier mobility process [1,2], provides excellent heat transfer and sensitivity [3,4], changes the morphologies and stability [5,6], and reveals the best mechanical, spectral, dynamic, and refractive characteristics [7,8,9,10,11,12,13,14,15]. Wide types and concentration range of nano-objects (nanoparticles, nanotubes, nanocrystals) are used to dope the polymer matrices, such as: fullerenes C_60_ and C_70_ [16,17,18,19,20], carbon nanotubes and nanofibers [21,22,23,24,25], quantum dots [26,27,28,29,30,31], reduced graphene oxides [32,33,34,35], MoS_2_, MoTe_2_, MoSi_2_, WS_2_, Au, Ag, TiO_2_, ZnO, CoFe_2_O_4_, etc. nanoparticles [36,37,38,39,40,41,42,43,44,45]. The doping process can improve the properties of polymer-based and hybrid composites.

Due to the effective use of pure and doped polymer materials in general optoelectronics, especially in laser technology, displays, and biomedicine, it is important to choose the scheme to test the main organic materials parameters. As has been shown in a paper [15], refractive property changes can be considered as the main parameter to indicate the improvement of organic materials’ physical–chemical features. It should be remarked that to reveal the change in refractivity, materials can be treated via the application of third harmonic generation [46,47], Z-scanning [48,49,50] set-up, and four-wave mixing technique (FWMT) [51,52,53,54]. The third harmonic generation scheme treats materials in a nonreversible mode. Thus, some cavities and other destructions can be detected even after the irradiation of the materials in a single mode according to this scheme. The Z-scanning set-up does not permit treatment of materials at the different spatial frequencies in order to reveal changes in refractivity over a broad range of energy and across different parts of the irradiated area. Moreover, it does not permit to separate the diffusion and drift mechanisms for the charge-carrier moving. Thus, the scheme of four-wave mixing of the laser beam has some advantages over the first two mentioned schemes. Choosing the appropriate method is essential to accurately assess key parameters of organic materials.

In a study [51], polythiophene films were treated using four-wave mixing techniques. The refractive parameters of these polymers were activated with significant advantage. In a paper [52], two-beam coupling and four-wave mixing measurements were presented, and their importance in the characterization of the photorefractive properties of the materials was explained. The physics of the photorefractive effect in polymers is discussed with emphasis placed on the differences compared with the traditional inorganic photorefractive crystals. In particular, the orientational enhancement mechanism, which is believed to be responsible for the high performance of most of the low-glass-transition-temperature systems, is discussed in detail. In manuscript [53], the FWMT scheme was used to study the bulk polymers based on poly-(methyl methacrylate) and poly-(vinyl toluene). A coherent signal resulting from the optical four-wave mixing is obtained with both the electronic and the Raman-resonant parts. The magnitude of the third-order optical susceptibility tensor of these polymers is experimentally estimated. Moreover, significant enhancement of the total susceptibility by the Raman resonances is observed for the poly-(vinyl toluene) system. In publication [54], the third-order optical nonlinearity of a series of new conjugated silicon-ethynylene polymers, poly(aryleneethynylenesilylene)s, was studied using the degenerate four-wave mixing technique. The fast nonlinear optical susceptibilities of the polymers containing various groups were determined for solutions in chloroform, tetrahydrofuran and toluene. The electronic and nuclear contributions of the χ^(3)^ susceptibility and the thermal nonlinearity of the solutions were separated.

It should be noticed that all of the applied schemes can reveal the refractive coefficients of the materials, which are responsible for the basic features of the materials. However, it is the four-wave mixing scheme of light beams, especially its modification in the holographic version, that allows non-destructive testing of the materials. This approach also allows determining the refraction parameters at different spatial frequencies in a reversible recording mode. It should be mentioned that the reversible recording is based on a change in the refractive index of the medium, i.e., reversible recording of information is implemented on the changes (variations) in the refractive index Δ*n*. These variations can be caused, in particular, by the high-frequency Kerr effect. The principles and methods of the four-wave mixing techniques have been well explained in papers [55,56,57].

We used the holographic set-up to reveal the laser-induced refractive index, to measure the diffraction efficiencies at low and high spatial frequencies, and to estimate the nonlinear refraction *n*_2_ and cubic nonlinearity χ^(3)^. This, among other things, makes it possible to separate the mechanism of the drift and diffusion of the charge carriers in doped organic materials. More efficient results were shown and discussed in papers [15,58,59,60]. It was proposed to take into account the large contribution of the intermolecular charge transfer complex formation mechanism to the induced refractive index, and also showed the dramatically loss of energy due to the diffraction on the lattice of a changing refractive index, which is important for the optical restriction process.

In this study, we investigated various organic thin-film structures including polyimides, COANPs, NPPs, and PNPs, which were sensitized with different types of nano-objects such as fullerenes, carbon nanotubes, quantum dots, shungite, and reduced graphene oxides. These structures were analyzed using the holographic technique under the Raman–Nath diffraction conditions

## 2. Materials and Methods

The photorefractive characteristics have been studied using the four-wave mixing technique, verified in the holographic version, which has been analogous to the set-up presented in papers [15,58]. The second harmonic of a pulsed Nd-laser at the wavelength λ = 532 nm was used. The laser energy density *W* was chosen in the range of 0.06–0.7 J × cm^−2^. The range of the spatial frequency Λ from 90(100) to 150 mm^−1^ was applied. The nanosecond laser regime with a pulse width τ of 10–20 ns was used. The amplitude-phase thin gratings were recorded under the Raman–Nath diffraction mode, where the ratio between the inverse of the spatial frequency Λ and the thickness of the thin-film medium *d* should be as follows: Λ^−1^ ≥ *d*. It should be noticed that thin-film polarizers have been used in this scheme operation, which have been developed in Kamanina’s lab. The diffraction efficiencies were measured, and after that, the change in the laser-induced refractive index was estimated, and the nonlinear refractive coefficient and cubic nonlinearity were calculated.

The thin films of different polymer-based composites (polyimides, 2-cyclooctylarnino-5-nitropyridine (COANP), N-(4-nitrophenyl)-(L)-prolinol (NPP), 2-(n-Prolinol)-5-nitropyridine (PNP), sensitized with different nano-objects at varied concentration *k* were prepared. Fullerenes, shungite, and reduced graphene oxides concentrations were varied in the range of 0.1–5.0 wt.%. The concentration of carbon nanotubes was in the range of 0.01–0.1 wt.%, that of the carbon nanofibers was close to 0.1 wt.%, and the content of quantum dots was close to 0.003–0.03 wt.%. The fullerenes were purchased from Alfa Aesar (Kurlsruhe, Germany). Double-walled carbon nanotube powder XNM-HP-11050 was received from XinNano Materials, Inc. (Taoyuan, Taiwan). Shungite structures were produced by Karelian Research Centre RAS. Carbon fibers have been sensitized by VlSU (Vladimir, Russia), and by Borescov Institute of catalysis SB RAS (Novosibirsk, Russia). The reduced graphene oxides nanoparticles were presented by the “NanoTechCenter” (Tambov, Russia). 

It should be mentioned that the role of the sensitization process of the polymer by different nanoparticles, which can increase the polarizability of the matrix compounds, increase the dipole moment, and change their refractive properties, was discussed in detail in paper [15].

It should be noticed that in our experiments, the nano-objects-doped organic films were prepared by spin-coating of 3–6.5% solutions of photosensitive components dissolved in 1,1,2,2,-tetrachloroetane and placed on glass substrates. It should be remarked that tetrachloroethane is an active solvent for the preparation of the doped films, as it dissolves most organic compositions and fullerenes as well [61]. The sensitized organic films had a thickness close to 2–5 μm. Indeed, some laboratory instruments, such as Visible (VIS) and near-infrared Fourier-spectrophotometers, atom force microscope (AFM), and scanning electron microscope (SEM), were used to test the homogeneity of the films developed. The thin films most suitable for the laser experiments were selected from among a large number of the rejected ones. 

## 3. Results and Discussion

The basic results correlated with the treatment of the polyimide-based composites are shown in Table 1 [15,60,62,63,64,65,66,67]. The following parameters are named in Table 1: *k*—concentration; *W*—laser energy density; Λ—spatial frequency; τ—laser pulse width; Δ*n*_i_—laser-induced change of the refractive index.

Let us discuss the results shown in Table 1.

Indeed, some data presented in Table 1 are compared with the results obtained before for these types of the materials. Indeed, we have previously studied these compounds under the different concentrations of doping agents, at different spatial frequencies, and across a range of energy densities. However, many novel data have been added to the table’s body and are indicated as c.d. (column Refs). It should be remarked that the holographic recording scheme requires good alignment and data processing; therefore, obtaining new data using this scheme expands our knowledge and complements the database on such materials.

It is important to note that different mechanisms responsible for refractivity changes, especially those resulting in laser-induced refractive index changes at different spatial frequencies, need to be carefully considered. It seems that the mechanism coincided with the charge drift process dominates at smaller spatial frequencies (at larger grating period and longer distance for the pathway of the charge); however, the diffusion process dominates at larger spatial frequencies (with a smaller grating period and shorter distance for the pathway of the charge). It should be taken into account that in our experiment, the charge drift process can be activated by the electric part of the light (laser) wave without the direct electric filed (bias voltage). The materials are treated in the passive regime. Thus, the experiments are conducted when the electrical component of the laser wave is larger than the intra-atomic electric field (the last one is proportional to the charge of the electron and inversely proportional to the square of the Bohr radius).

Based on the laser-induced change in the refractive index, the nonlinear refraction coefficient, *n*_2_, and cubic nonlinearity, χ^(3)^, can be calculated using equations [55]: (1)$$n_{2}=\frac{\Delta n_{i}}{I}$$
(2)$$\chi^{\left(3\right)}=\frac{n_{2}n_{0}c}{16\pi^{2}}$$

Here Δ*n*_i_ is the laser-induced change in the refractive index, *I* is the intensity of the laser beam, *n*_0_ is the linear refractive index of the system, and *c* is the light velocity. Thus, the nonlinear refraction coefficient, *n*_2_, and the third order susceptibility, χ^(3)^, can be found to be close to the values, respectively: ~10^−8^–10^−7^ cm^2^kW^−1^ and ~10^−10^–10^−9^ esu for thin conjugated films of the doped polyimide structures. These values of the nonlinear optical coefficients for nanostructured organics systems, shown here and based on the previously estimated two photon absorption coefficient β [68], are essentially larger than those obtained for bulk inorganic materials, such as LiNbO_3_, KDP, and DKDP crystal structures [69,70,71]. Thus, taking into account the importance of the refractive parameters of organic sensitized thin-film materials, we can say that they have an advantage over the volumetric inorganic ones due to their design and the simplicity of embedding into any optoelectronic circuits.

Data shown in Table 2 [62,72,73,74,75] present the refractive properties of the COANP, NPP, and PNP materials doped with effective fullerene nanoparticles. It should be remarked that these types of materials in their not-doped (pure) form have been studied in detail in papers [76,77,78,79,80,81,82,83]. In paper [77], for example, COANP was doped with the dyes TCNQ (7,7,8,8,-tetracyanoquinodimethane) and treated with the laser at 675 nm in order to activate the nonlinearity. At the energy density of 2.2 W × cm^−2^, the induced change of the refractive index was revealed to be close to 2 × 10^−5^. We proposed doping the COANP structure with the fullerenes [84,85], which have a larger electron affinity energy than that of the intramolecular acceptor. We testified that the spectral features were dramatically modified in the doped COANP and explained the effect via the dipole–dipole interaction in the studied organics. This led to the subsequent instigation of these materials using a holographic approach [67,72,73,74,75].

It is interesting to note that the materials based on pyridine components (see Table 2 data) were irradiated with a laser with a higher energy density than shown for the polyimide systems (see Table 1 data). It should be borne in mind here that the intramolecular acceptor fragment of the pyridine structures (~0.45 eV [86,87]) is significantly smaller than the intramolecular fragment of a polyimide molecule (~1.14–1.4 eV [88,89]). Most likely, during the formation of the intermolecular charge transfer complex, the contribution of the intramolecular process significantly weakens for the pyridine components; their activation requires a higher value of the incident laser energy. It should be further considered that the electron affinity of both fullerenes is close to 2.6–2.7 eV [90,91,92] which is twice as large as that of an intramolecular acceptor fragment of the polyimide, and four times larger than that for COANP. Therefore, the dopants evidently dominate over the intramolecular acceptor fragments. The formed intermolecular charge transfer process between dopants and polyimides and COANP donor fragments enhances the phototransfer of the charge in these systems resulting in the observed increase in the refractive index.

Thus, we are dealing with a significant charge shift to the intermolecular acceptor, which causes an increase in the anisotropy of the system, its polarizability, and modification of the refractive parameters (for example, laser-induced change of the refractive index increases from ~10^−5^ (pure materials) up to 10^−3^ (doped materials)). Thus, such an intermolecular charge transfer process can be ideally modeled as the creation of a kind of intramolecular Janus particle, which can significantly simplify the understanding of this effect of the refraction increase. Indeed, it can be supported by the direct estimation of the studied organics doped with Janus nanoparticles, but this idea will be tested in the following further experiments. The extended version of the possible dominant role of the intermolecular charge transfer complex formation with the nanoparticles added shown previously in papers [14,15,93], is presented in Figure 1. The visualization of the amplitude-phase holographic grating is shown as well. The arrow on the left side of the Figure 1 shows the change in the path of the charge carrier transfer during intermolecular complexation with the introduction of the nanoparticles. These nanostructures are shown above (above the line) and below (below the line). This is a pretty big search for nanoobjects.

One can see that the 3D local media can be created (right part of Figure 1) due to the diffraction of the laser beam at the nanostructured fragment of the materials. It can provoke, from one side, the increase of the density of the recorded information; from the other side, it presents the energy losses via this diffraction. Different periods of the grating in the *x* and *y* axis are established. 

The results shown in Table 1 and Table 2 are connected with the optical limiting ability of the materials studied. It should be noticed that the increase of the refractive parameters predicts including the additional mechanism in the optical limiting process explanation. Really, energy losses via the diffraction on the structured materials can decrease the intensity (energy density) transferred through the nonlinear optical media.

Once again, the range of physical mechanisms responsible for nonlinear absorption of optical radiation—such as inverse saturated absorption [12,94,95], absorption by charge transfer complexes via the Förster mechanism [2,13,96], absorption by free carriers, and absorption of impurities [97,98]—is further expanded by the impact of energy loss due to diffraction in nanostructured materials. The dramatic change in the laser-induced refractive index can effectively support this evidence.

Moreover, it should be remarked that the refractive index change is successfully correlated with the photoconductive and dynamic properties of the structured organic thin films characteristics. Some of this evidence has been shown in papers [64,99], some of these should be considered and supported in future. Furthermore, the results obtained are correlated with the data previously found for the bio-doped organics, shown in paper [100].

## 4. Conclusions

Analyzing the results shown one can postulate the following.

The refractive properties of organic materials, especially nanostructured ones, can be effectively studied using a holographic technique. This approach permits finding the refractive index change at different spatial frequencies and can separate the mechanisms responsible for the nonlinear optical feature modifications. Novel data have been obtained for the organics studied, when the fullerenes, shungites, nanotubes, and the graphene oxide nanoparticles have been added.The laser-induced refractive index of the nanostructured organic materials is larger than that obtained for pure materials. Really, the values of the induced refractive index are two orders of magnitude higher than those obtained for pure films. It can be explained by the formation of the intermolecular charge transfer process that can add an additional mechanism to the laser-matter interaction. The polarizability of the treated system, indicated via an increase of the local volume polarizability χ^(3)^, can increase as well.The laser-induced refractive index of the nanostructured organic materials obtained at the Raman–Nath diffraction condition can predict larger nonlinear optical coefficients, which are close to or larger than the same value established by different scientific teams for the classical volumetric inorganic structures. The estimated nonlinear refraction coefficient, *n*_2_, and the third order susceptibility, χ^(3)^, can be found to be close to the values, respectively: ~10^−8^–10^−7^ cm^2^kW^−1^ and ~10^−10^–10^−9^ esu for thin conjugated films of the doped structures.The obtained laser-induced refractive index of the nanostructured organic materials can reveal the existing novel mechanics of the attenuation of the laser beam, namely, energy loss via diffraction, which can be added to the optical limiting process explanation. Moreover, the 3D local media can be created that can be useful to create devices with high-density recording of optical information.Among other processes, doping organics with different nano-objects can be considered as the creation of the Janus particles due to a drastic shift (displacement) of charge from the intramolecular donor fragment to the intermolecular acceptors. It is first proposed in this research to form the Janus nanoparticles via the displacement of charge in the doped organics. This shift of charge depends on the relation between the intramolecular electron affinity energy and the electron affinity energy of the particles, used for the doping.The studied nanostructured materials can be considered for fundamental discussion in the material science area and can be proposed for a wide range of applications: in holography, display, modulation, conversion, and limiting of the laser beam, sensors and biomedicine.The results shown can be involved in student education in universities due to the reason that the procedure to develop the thin-film organics structures and test them via a holographic set-up is a well-visualized process.

## Figures and Tables

**Figure 1 polymers-16-02645-f001:**
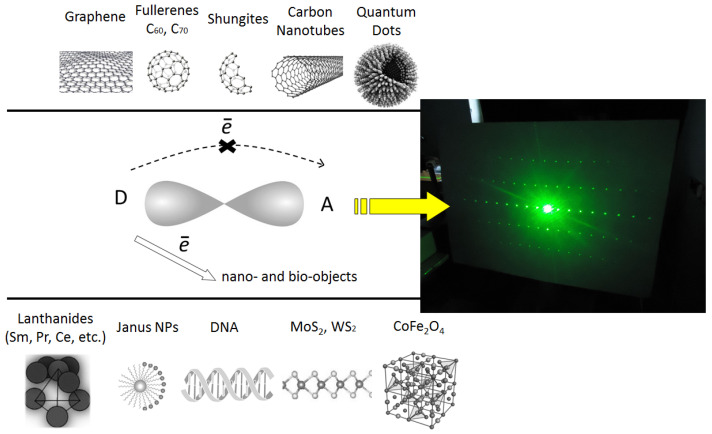
The proposed extended scheme of intermolecular charge transfer using organic conjugated materials, where the introduced intermolecular doping nano-object has an electron affinity significantly greater than the intramolecular acceptor of the matrix system.

**Table 1 polymers-16-02645-t001:** Laser-induced change of the refractive index of sensitized polyimides.

Structure Studied	*k* wt.%	*W*, J × cm^−2^	Λ, mm^−1^	τ, ns	Δ*n*_i_	Refs.
Pure PI	0	0.6	90	20	10^−4^–10^−5^	[62]
PI+malachite green	0.2	0.5–0.6	90–100	10–20	2.87 × 10^−4^	[63]
PI+QDs CdSe(ZnS)	0.003	0.2–0.3	90–100	10	2.0 × 10^−3^	[64]
PI+QDs CdSe(ZnS)	0.03	0.2	90–100	10	2.2 × 10^−3^	[15]
PI+shungite	0.1	0.6	100	10	3.6 × 10^−3^	[15]
PI+shungite	0.15	0.5	100	10	3.7 × 10^−3^	c.d. *
PI+shungite	0.2	0.063–0.1	150	10	(3.8–5.3) × 10^−3^	[65]
PI+C_60_	0.1	0.5	100	10	3.5 × 10^−3^	c.d.
PI+C_60_	0.2	0.5–0.6	90	10–20	4.2 × 10^−3^	[62]
PI+(C_60_+C_70_)	0.2	0.5	100	10	4.4 × 10^−3^	c.d.
PI+C_70_	0.1	0.5	100	10	4.2×10^−3^	c.d.
PI+C_70_	0.2	0.6	90	10–20	4.68 × 10^−3^	[62]
PI+nanotubes	0.1	0.5–0.8	90	10–20	5.7 × 10^−3^	[62]
PI+nanotubes	0.05	0.3	150	10	4.5 × 10^−3^	[66]
PI+nanotubes	0.1	0.3	150	10	5.5 × 10^−3^	[66]
PI+nanotubes	0.15	0.5	100	10	5.6 × 10^−3^	c.d.
PI+DWCNT powder	0.1	0.063–0.1	100	10	9.4 × 10^−3^	[60]
PI+DWCNT powder	0.1	0.063–0.1	150	10	7.0 × 10^−3^	[60]
PI+carbon nanofibers	0.1	0.6	90–100	10	11.7 × 10^−3^	[65]
PI+carbon nanofibers	0.1	0.3–0.6	150	10	11.2 × 10^−3^	[60]
PI+carbon nanofibers	0.1	0.1–0.3	90–100	10	12.0 × 10^−3^	[65]
PI+carbon nanofibers	0.1	0.1	90	10	15.2 × 10^−3^	[60]
PI+RGrO	0.1	0.2	100	10	3.4 × 10^–3^	[67]
PI+RGrO	0.15	0.2	100	10	3.5 × 10^–3^	c.d.
PI+RGrO	0.1	0.2	150	10	3.1 × 10^–3^	[15]

* c.d.—current data.

**Table 2 polymers-16-02645-t002:** Laser-induced change of the refractive index of sensitized COANP, etc., systems.

Structure Studied	*k* wt.%	*W*, J × cm^−2^	Λ, mm^−1^	τ, ns	Δ*n*_i_	Refs.
Pure COANP	0.0	0.9	90	20	~10^−5^	[62,72]
COANP+C_60_	2.0	0.7	100	10	5.2 × 10^−3^	c.d. *
COANP+C_60_	5.0	0.9	90–100	10–20	6.21 × 10^−3^	[62,73]
COANP+C_70_	0.5	0.6	100	10–20	(4.5–5.1) × 10^−3^	[74]
COANP+C_70_	2.0	0.7	100	10	5.4 × 10^−3^	c.d.
COANP+C_70_	5.0	0.9	90–100	10–20	6.89 × 10^−3^	[62,73]
Pure NPP	0.0	0.3	100	20	0.65 × 10^−3^	[75]
NPP+C_60_	1.0	0.3	100	20	1.65 × 10^−3^	[75]
NPP+C_70_	1.0	0.3	100	20	1.20 × 10^−3^	[75]
Pure NPP	0.0	0.7	100	10	~10^−5^	c.d.
NPP+C_60_	1.0	0.7	100	10	4.2 × 10^−3^	c.d.
NPP+C_70_	1.0	0.7	100	10	4.5 × 10^−3^	c.d.
Pure PNP	0.0	0.3	100	20	-	[75]
PNP+C_60_	1.0	0.3	100	20	0.8 × 10^−3^	[75]
Pure PNP	0.0	0.7	100	10	~10^−5^	c.d.
PNP+(C_60_+C_70_)	1.0	0.7	100	10	3.8 × 10^−3^	c.d.
PNP+C_60_	1.0	0.7	100	10	4.0 × 10^−3^	c.d.
PNP+C_70_	1.0	0.7	100	10	4.3 × 10^−3^	c.d.

* c.d.—current data.

## Data Availability

The original contributions presented in the study are included in the article/Appendix A, further inquiries can be directed to the corresponding author/s.

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
