# Peer review of "Advances of the Holographic Technique to Test the Basic Properties of the Thin-Film Organics: Refractivity Change and Novel Mechanism of the Nonlinear Attenuation Prediction"

_polymers, 2024, doi:10.3390/polym16182645_

Round 1
Reviewer 1 Report
Comments and Suggestions for Authors
In this work, the thin films of different polymer-based composites sensitized with different nanoobjects with the varied content have been prepared. The large types of the organic materials including doped ones are shown via their test using the holographic set-up. Some detailed comments:
*Abbreviations should not be used for terms appearing for the first time in the abstract and in the text, e.g., COANP, NPP, PNP.
*The authors mention that, in the current study the large types of the organic materials (polyimides, COANP, NPP, PNP), including doped ones are shown via their test using the holographic set-up. This seems to describe only what the article does. What is the innovation of this article?
*How is the film prepared? What is the source of the raw materials needed?
*The article reads like an integration of a literature review with the test results obtained by the authors, with only two tables for the main data. As a scientific paper, the article is not well researched.
*Conclusions-I think all these conclusions can be obtained by summarizing the existing literature. What is the contribution of the authors in conducting the experiment? Are there new conclusions that can be drawn from the authors' experimental data?
In conclusion, this article is recommended to be major revised.
Comments on the Quality of English LanguageMinor editing of English language required.
Author Response
Comments and Suggestions for Authors
In this work, the thin films of different polymer-based composites sensitized with different nanoobjects with the varied content have been prepared. The large types of the organic materials including doped ones are shown via their test using the holographic set-up. Some detailed comments:
*Abbreviations should not be used for terms appearing for the first time in the abstract and in the text, e.g., COANP, NPP, PNP.
*The authors mention that, in the current study the large types of the organic materials (polyimides, COANP, NPP, PNP), including doped ones are shown via their test using the holographic set-up. This seems to describe only what the article does. What is the innovation of this article?
*How is the film prepared? What is the source of the raw materials needed?
*The article reads like an integration of a literature review with the test results obtained by the authors, with only two tables for the main data. As a scientific paper, the article is not well researched.
*Conclusions-I think all these conclusions can be obtained by summarizing the existing literature. What is the contribution of the authors in conducting the experiment? Are there new conclusions that can be drawn from the authors' experimental data?
In conclusion, this article is recommended to be major revised.
Comments on the Quality of English Language
Minor editing of English language required.
Dear Reviewer!
Thank you very much for your comments to this paper.
Please see my answers:
1). Yes! Thank you. I am agreeing with that.
I wrote the full name of the substances (COANP, NPP, PNP as 2-cyclooctylarnino-5-nitropyridine, N-(4-nitrophenyl)-(L)-prolinol, 2-(n-Prolinol)-5-nitropyridine) in the theses that we met for the first time. You are right; one cannot immediately write an abbreviation without specifying the chemical structure. The additional words have been collared with yellow.
2). Sorry, maybe I have not explained well before. Please see data from Table 1. I have indicated as c.d.*- current data.
Thus, so many novel data have been added in the text body. Indeed, we have studied before these compounds under the different content of the doping agents, at the different spatial frequencies and different range of the energy densities. But, I have added absolutely new results. They can extend the parameters of the induced refractive index and extend the mechanism responsible for the writing holographic grating under the Raman-Nath conditions.
The holographic recording scheme requires good alignment and data processing; therefore, obtaining new data using this scheme expands our knowledge and complements the database on such materials. I hope that I am answering to your second comments.
Please the paragraph (in Results and discussion section), which I have added before Table 1 in the text body.
Indeed, some data presented in Table 1 are compared with the results obtained before for these types of the materials. Indeed, we have studied before these compounds under the different content of the doping agents, at the different spatial frequencies and different range of the energy densities. But, so many novel data have been added in the table’s body and indicated as c.d. (column Refs). It should be remarked that the holographic recording scheme requires good alignment and data processing; therefore, obtaining new data using this scheme expands our knowledge and complements the database on such materials.
3). About the film preparation.
Thank you! I have added the sentence explained the films preparation. Please see the paragraph (in Materials and Method section) collared with yellow.
It should be noticed that in our experiments, the nanoobjects-doped organic films have been prepared by spin-coating of 3–6.5% solutions of a photosensitive components solved in 1,1,2,2,-tetrachloroetane and placed on a glass substrate.
4). About your comments that the article reads like an integration of a literature review…
By analyzing the new data obtained, I tried to show that this experimental technique allows us to adequately compare old data with new ones, reveals the features of estimating changes in the refractive index at small and large spatial frequencies, and shows the measurement of the refraction, as a basic parameter, when varying the introduced nanoobjects. It is researcher’s local view. Maybe it is not traditional one, but, sorry, it can be postulated.
5). About Conclusion part. Yes, Thank you, I am agreeing with you and a little bit revised Conclusion section.
6). English correction needed… Thank you! I have corrected my English, Sorry, at the present time we have no so large practices in English. But, I try to improve the text of the paper.
All corrections (including articles) have been collared with yellow.
I would like to note that your comments are improved my paper, it is true. Thanks a lot once again!
Best Regards,
Natalia Kamanina
=======================================
Natalia V. Kamanina (Prof., Dr.Sci., PhD)
Head of the lab for Photophysics of media with nanoobjects
Vavilov State Optical Institute
Kadetskaya Liniya V.O., dom.5, korpus 2,
St.- Petersburg, 199053, Russia
Professor of the St.-Petersburg Electrotechnical University (“LETI”),
Part-time Leading Researcher at Nuclear Physics Institute (Gatchina)
Job phone: +7 (812) 327-00-95
Fax: +7 (812) 331-75-58 (for N.V.Kamanina)
e-mail: nvkamanina@mail.ru
Lab_cite: sites.google.com/view/photophysics-lab
https://publons.com/researcher/1696479/natalia-kamanina/
https://sciprofiles.com/news-feed
http://rusnor.org/network/webinars/10203.htm
http://www.npkgoi.ru/?module=articles&c=profil&b=7
http://www.nanometer.ru/2007/08/09/liquid_crystal_3905.html
https://etu.ru/ru/fakultety/fakultet-elektroniki/sostav-fakulteta/kafedra-fotoniki/sostav-kafedry
=======================================

Reviewer 2 Report
Comments and Suggestions for Authors
This paper presents an analysis of thin-film organic structures sensitized with some nano-objects using holographic techniques under Raman-Nath diffraction conditions. The paper suggests that pulsed laser irradiation significantly increases the laser-induced refractive index in these materials, with nonlinearities comparable to or greater than those in volumetric inorganic crystals. Intermolecular charge transfer complexes and energy losses due to diffraction are key factors in this increased refractivity and nonlinear attenuation.
The article provides a thorough and detailed presentation of the previous literature, which is one of the strengths of the manuscript. The paper also discusses all the results observed with reference to the knowledge already conveyed in previous research work. This is greatly appreciated from a scientific point of view. The paper certainly has merit, but before being published it must undergo a thorough revision on several points (please see the attachd file).

The English throughout the manuscript is very heavy and difficult to follow. A much easier-to-understand writing style should be chosen by the paper's author. In several compartments of the manuscript, the author must divide his long sentences into several easy-to-follow phrases for the readers. The very essence of a scientific publication is the sharing of knowledge. The author is invited to revise the manuscript by a English specialist, a native English speaker or by editing services such as those offered by MDPI.
Author Response
Comments and Suggestions for Authors
This paper presents an analysis of thin-film organic structures sensitized with some nano-objects using holographic techniques under Raman-Nath diffraction conditions. The paper suggests that pulsed laser irradiation significantly increases the laser-induced refractive index in these materials, with nonlinearities comparable to or greater than those in volumetric inorganic crystals. Intermolecular charge transfer complexes and energy losses due to diffraction are key factors in this increased refractivity and nonlinear attenuation.
The article provides a thorough and detailed presentation of the previous literature, which is one of the strengths of the manuscript. The paper also discusses all the results observed with reference to the knowledge already conveyed in previous research work. This is greatly appreciated from a scientific point of view. The paper certainly has merit, but before being published it must undergo a thorough revision on several points (please see the attachd file).
Comments on the Quality of English Language
The English throughout the manuscript is very heavy and difficult to follow. A much easier-to-understand writing style should be chosen by the paper's author. In several compartments of the manuscript, the author must divide his long sentences into several easy-to-follow phrases for the readers. The very essence of a scientific publication is the sharing of knowledge. The author is invited to revise the manuscript by a English specialist, a native English speaker or by editing services such as those offered by MDPI.
Dear Reviewer!
Thank you very much for your comments to this paper!
Please see my answers:
General answers about the previously obtained data and novel ones.
As for my local opinion, so many novel data have been added in the text body. Indeed, we have studied before these compounds under the different content of the doping agents, at the different spatial frequencies and different range of the energy densities. But, I have added absolutely new results. They can extend the parameters of the induced refractive index and extend the mechanism responsible for the writing holographic grating under the Raman-Nath conditions.
The holographic recording scheme requires good alignment and data processing; therefore, obtaining new data using this scheme expands our knowledge and complements the database on such materials. I hope that I am answering to your comments as: “… The paper also discusses all the results observed with reference to the knowledge already conveyed in previous research work…”.
Next, I will answer your questions according .pdf-file. Only I don't have a line division in my file, I might make a mistake in the order, I'm sorry.
1). Lines 14-17: you could have expressed yourself much more easily, for example:
In this study, we investigated various organic thin-film structures including polyimides, COANPs, NPPs, and PNPs, which were sensitized with different types of nano-objects such as fullerenes, carbon nanotubes, quantum dots, shungite, and reduced graphene oxides. These structures were analyzed using the holographic technique under Raman-Nath diffraction conditions.
Thanks a lot for your help. It is really the best! I have included this paragraph in the text body, as the last paragraph of Introduction section. It is collared with green.
In this study, we investigated various organic thin-film structures including polyimides, COANPs, NPPs, and PNPs, which were sensitized with different types of nano-objects such as fullerenes, carbon nanotubes, quantum dots, shungite, and reduced graphene oxides. These structures were analyzed using the holographic technique under Raman-Nath diffraction conditions
2). Line 44: what do you mean by ‘’ it is important to choose the scheme to testify the main organic materials parameters’’. Did you mean: Choosing the appropriate method is essential to accurately assess key parameters of organic materials ?
Yes, thank you once again. This sentence is the best! I have added in the Introduction part this sentence:
Choosing the appropriate method is essential to accurately assess key parameters of organic materials.
3). Lines 79-83: You sentence is too long and difficult to follow Please divide your long sentences in shorter ones to make them easier to understand. Did you mean: However, it is the four-wave mixing scheme of light beams, especially its modification in the holographic version, that allows non-destructive testing of materials. This approach also allows to determine the refraction parameters at different spatial frequencies in a reversible recording mode.
Thank you! You correct the understanding of the paragraph!
I have replace the sentence: “But, it is namely the scheme of the four-wave mixing of the light beams, its modification in the holographic version, which makes it possible to implement a non-destructive testing mode for the materials, as well as to determine the refractive parameters at the different spatial frequencies of the recording information in the reversible mode” for the best variant, you proposed:
However, it is the four-wave mixing scheme of light beams, especially its modification in the holographic version, that allows non-destructive testing of materials. This approach also allows to determine the refraction parameters at different spatial frequencies in a reversible recording mode.
4). Line 87: Did you mean: The principles and method of the missing four waves techniques have been well explained in [55-57]?
Yes, perhaps I should have added the Kogelnik’ book, but then it would have become necessary to figure out where the record of the rectangular lattice is, where the sinusoidal structure is. This would complicate the text and become incomprehensible to some readers. Sorry, I have not the Kogelnic book.
5). Lines 98-99: Did you mean: In the present study, many types of organic materials (polyimides, COANP, NPP, PNP), including doped materials, were tested using a holographic device ?
Thank you! I have used the set-up, the scheme, the technique, the device…as the approach to test the materials via application of the holographic recording method.
6). Lines 144-146: Please revise your sentence. Did you mean: It is important to note that the different mechanisms responsible for refractivity changes, especially those resulting in laser-induced refractive index changes at different spatial frequencies, need to be carefully considered ?
Thank you for you correction! I have replace the sentence: “It should be noticed that the different mechanisms responsible for the refractivity change, for the laser induced refractive index modifications at the different spatial frequencies should be drawn into attention” for your proposed variant:
It is important to note that the different mechanisms responsible for refractivity changes, especially those resulting in laser-induced refractive index changes at different spatial frequencies, need to be carefully considered.
7). Line 230: ‘’Once again, consequently, the variety…’’. Please revise the whole sentence. Here is a suggestions if this goes along with what you meant : Once again, the range of physical mechanisms responsible for nonlinear absorption of optical radiation - such as inverse saturated absorption [12,94,95], absorption by charge transfer complexes via the Förster mechanism [2,13,96], absorption by free carriers and absorption of impurities [97,98] - is further expanded by the impact of energy loss due to diffraction in nanostructured materials.
Thank you!!!
I have replaced the sentence: Once again, consequently, the variety of the physical mechanisms responsible for the nonlinear absorption of the optical radiation, such as: reverse saturated absorption [12,94,95], absorption of the charge transfer complexes and due to the Förster mechanism [2,13,96], absorption on the free carriers and impurity absorption [97,98], etc., is supplemented by a change in the absorption due to the energy loss due to the diffraction in the nanostructured materials for the proposed variant from you:
Once again, the range of physical mechanisms responsible for nonlinear absorption of optical radiation - such as inverse saturated absorption [12,94,95], absorption by charge transfer complexes via the Förster mechanism [2,13,96], absorption by free carriers and absorption of impurities [97,98] - is further expanded by the impact of energy loss due to diffraction in nanostructured materials.
8).
- Lines 49 and 66: what does FWMT stands for? The same thing for VIS, AFM, SEM. It may seem obvious to you, but you really need to define them or present a table of acronyms at the end of your manuscript.
Well. I have revealed the abbreviations. They are colored with green in the text body.
9).
- Line 126: what does ‘’content k’’ stands for? The journal ''Polymers'' is aimed at a wide range of readers interested in polymeric materials. Please make your manuscript clear to all readers of your paper
- I have replaced the term “content, k” on concentration k.
10).
Lines 133 -136: The author cites several techniques used (SEM, AFM, VIS), but no results are presented. The author also explains that the thin films most suitable for the laser experiments were selected from many rejected ones. Explain how and on what exact criteria. Once again, it may seem obvious to you, but don't leave your readers in any doubt. Enrich your manuscript with at least more images of the samples you've chosen, and detailed results of your work.
Thank you for your comments. But, these instruments we have always used. I would not like to duplicate the same data that has already been demonstrated in the previously shown works.
11).
Lines 138-139: It's not at all clear from the sentence. But from the content of the table, the reader should understand that the author is comparing his results with those of previous works. Explain clearly what you are presenting in tables 1 and 2.
- Please define clearly the parameters presented in both tables 1 and 2. Some of the parameters in Table 1 are defined later in the text on page 5. The reader must search back and forth to find them. Present them below your table or in the text, the first time you talk about Table 1.
Well. The novel results, compared with the previously shown are named as c.d. – current data.
About the parameters indicated in Tables. I am so sorry, it is my mistake. Please see the sentence, which I have placed before Table 1.
The following parameters are named in Table 1, such as: k – concentration; W – laser energy density; L - spatial frequency; τ – laser pulse width; Dni – laser-induced change of the refractive index.
12).
Section 4 (Conclusions): Before turning to your conclusions, it would be a good idea to wrap up your results
Yes, it is good recommendation. I have included the results in the Conclusion part.
13). English correction needed… Thank you! I have corrected my English, Sorry, at the present time we have no so large practices in English. But, I try to improve the text of the paper.
- Thank you so much. Your recommendation and corrections are improved the paper with good advantage. All added parts have been collared with green.
Best Regards,
Natalia Kamanina
=======================================
Natalia V. Kamanina (Prof., Dr.Sci., PhD)
Head of the lab for Photophysics of media with nanoobjects
Vavilov State Optical Institute
Kadetskaya Liniya V.O., dom.5, korpus 2,
St.- Petersburg, 199053, Russia
Professor of the St.-Petersburg Electrotechnical University (“LETI”),
Part-time Leading Researcher at Nuclear Physics Institute (Gatchina)
Job phone: +7 (812) 327-00-95
Fax: +7 (812) 331-75-58 (for N.V.Kamanina)
e-mail: nvkamanina@mail.ru
Lab_cite: sites.google.com/view/photophysics-lab
https://publons.com/researcher/1696479/natalia-kamanina/
https://sciprofiles.com/news-feed
http://rusnor.org/network/webinars/10203.htm
http://www.npkgoi.ru/?module=articles&c=profil&b=7
http://www.nanometer.ru/2007/08/09/liquid_crystal_3905.html
https://etu.ru/ru/fakultety/fakultet-elektroniki/sostav-fakulteta/kafedra-fotoniki/sostav-kafedry
=======================================

Round 2
Reviewer 2 Report
Comments and Suggestions for Authors
The author of the paper responded to all comments. In some cases, she has decided to integrate comments and suggestions. This is appreciated. In other cases, she explains her choice not to include them. In view of these explanations, these reasons are comprehensible. Again, the fact that it provides a comprehensive review of the previous literature is a strength of this paper. It examines all the observed results in relation to existing research, is also appreciated. This approach to presenting the data is valuable from a scientific point of view. Overall, the paper is clearly meritorious.
Comments on the Quality of English LanguageThe author seems to have made an effort to improve the English level of the manuscript. Let's hope this will suit the editorial policies of the journal.